# Generative Unfolding with Distribution Mapping

Anja Butter[1,2], Sascha Diefenbacher[3], Nathan Huetsch[1], Vinicius Mikuni[4],
Benjamin Nachman[3,5], Sofia Palacios Schweitzer[1] and Tilman Plehn[1,6]

**1** Institut für Theoretische Physik, Universität Heidelberg, Germany
**2** LPNHE, Sorbonne Université, Université Paris Cité, CNRS/IN2P3, Paris, France
**3** Physics Division, Lawrence Berkeley National Laboratory, Berkeley, USA
**4** National Energy Research Scientific Computing Center, Berkeley Lab, Berkeley 94720, USA
**5** Berkeley Institute for Data Science, University of California, Berkeley, CA 94720, USA
**6** Interdisciplinary Center for Scientific Computing (IWR), Universität Heidelberg, Germany

November 6, 2024

## Abstract

**Machine learning enables unbinned, highly-differential cross section measurements. A recent idea uses generative models to morph a starting simulation into the unfolded data. We show how to extend two morphing techniques, Schrödinger Bridges and Direct Diffusion, in order to ensure that the models learn the correct conditional probabilities. This brings distribution mapping to a similar level of accuracy as the state-of-the-art conditional generative unfolding methods. Numerical results are presented with a standard benchmark dataset of single jet substructure as well as for a new dataset describing a 22-dimensional phase space of $Z + 2$-jets.**

## 1  Introduction

Differential cross sections are the central currency of exchange in particle and nuclear physics. These objects connect theory with experiment, allowing for most of the data analysis performed at particle colliders. Simulations of collisions from the smallest distance scales up to the macroscopic size of detectors enable comparisons of synthetic and real data [1]. Comparing fully simulated and real events directly has a number of advantages, but it also requires access to the experimental data and prespecifying the physics questions and calculations. An alternative approach that trades off precision with versatility/longevity is *unfolding*, where the data are corrected for detector effects. In this way, the resulting differential cross sections are independent of their detector and can be interrogated with a number of questions and calculations, even those that were not known at the time of the measurement.

Classical unfolding methods act on histograms to produce binned, differential cross section measurements [2–5]. These tools have been used for a multitude of measurements over decades and have in turn been used for many downstream analyses. However, traditional approaches are fundamentally limited in scope. Recently, there have been a number of proposals to use modern machine learning (ML) to address these limitations [6, 7]. ML-unfolding is either based on reweighting [8–11] or generative networks [12–24], and it allows for unbinned measurements with a large number of input and output dimensions. This greatly increases the long-term utility of the measurements, as the cross section of new observables can be automatically extracted from the result. Experimental results with these tools are already being published, including measurements based on the OmniFold method [8, 9] from H1 [25–28], LHCb [29], CMS [30], STAR [31], and ATLAS [32]. These results are promising, but due to the ill-posed nature of the problem, it is essential to have alternative methods.

In this paper, we focus on generative unfolding, and in particular, on the key step of sampling from likely particle-level (gen-level) events given detector-level (reco-level) events*. One class of generative model-based approaches use distribution mapping, whereby the experimental events are morphed to match the corresponding gen-level events. By starting from the experimental events directly, the generative model only needs to move the events a little (assuming a precise detector), whereas other generative approaches need to map generic Gaussian random variables into the data distribution. Previously, two distribution mapping approaches were proposed, both based on conditional diffusion models [33]: one using Schrödinger Bridges (SBUnfold [18]) and one using Direct Diffusion (DiDi [23]). Previous work showed that these techniques showed excellent performance on the marginal distributions of the target cross sections, but they were not able to preserve the conditional distributions of the detector response. This could lead to a strong dependence on the gen-level simulation and is thus undesirable. The goal of this paper is to remedy this issue through conditioning [34]. Along the way, we introduce a new benchmark dataset, inspired by the recent ATLAS measurement [32], that can be used for distribution mapping as well as any unfolding method.

---

*A complete unfolding approach might include approaches to mitigate the dependence on the starting simulation [17] as well as acceptance effects [9].

This paper is organized as follows. Section 2 explains distribution mapping in detail, including a remedy to the challenge with current methods. This section provides important and useful derivations, but it can be skipped without interrupting the flow of the paper. Section 3 highlights the fix to distribution mapping methods with a simple, illustrative example. Next, the new methods are tested on a benchmark dataset of single jet substructure 4 and then we create a a new dataset describing a 22-dimensional phase space in $Z + 2$-jets at the Large Hadron Collider 5. This latter dataset combines jet substructure and kinematic information. For all applications, we provide a detailed discussion of the conditional Schrödinger Bridge and Direct Diffusion performance and a comparison with the state of the art in generative unfolding, a diffusion model using transformer layers [14, 23]. The paper ends with conclusions and outlook in Sec. 6.

## 2 Distribution Mapping

Diffusion networks offer the possibility of sampling a phase space distribution from any given distribution, not just from Gaussians or other standard distributions [18, 19, 23, 35]. Before we use this for unfolding, we review in some detail about how this feature can be realized for stochastic conditional sampling. The derivations presented in this section are not new and not necessary to understand the results for the simple model in Sec. 3 and for the physics examples in Secs. 4 and 5. Nevertheless, we use this opportunity to carefully describe the methods in a physics context and illustrate how score matching (Schrödinger Bridge) and velocity matching (Direct Diffusion) approaches unify.

### 2.1 Distribution to noise

We want to design a process $p(x, t)$ that transforms a general data distribution (at reco level) into a general latent distribution (at gen level),

$$p(x, t) = \begin{cases} p_{\text{data}}(x) & t = 0 \\ p_{\text{latent}}(x) & t = 1 . \end{cases} \tag{1}$$

This can be done using the stochastic differential equation (SDE)

$$dx = f(x, t) \, dt + g(t) \, dW . \tag{2}$$

Here $f(x, t)$ is the so-called drift coefficient, describing the deterministic part of the time evolution. The diffusion coefficient $g(t)$ describes the strength of the noising process, and $W$ is a standard Wiener process, $dW$ a noise infinitesimal. The connection between the evolving density in Eq.(1) and the trajectories in Eq.(2) is given by the Fokker-Planck equation (FPE)

$$\frac{\partial p(x, t)}{\partial t} = -\nabla_x [f(x, t) p(x, t)] + \frac{g(t)^2}{2} \nabla_x^2 p(x, t) . \tag{3}$$

For $g(t) = 0$ the SDE reduces to an ordinary differential equation (ODE), and the FPE to the usual continuity equation. Unlike ODEs, SDEs are not time-symmetric. This is because adding noise to a system is fundamentally different from removing noise. It can be shown that the time-reversal of Eq.(2) is given by another diffusion SDE [36],

$$dx = \left[ f(x, t) - g(t)^2 \, \nabla_x \log p(x, t) \right] dt + g(t) \, d\bar{W} . \tag{4}$$

where $d\bar{W}$ is the noise infinitesimal corresponding to the reverse Wiener process. The new and unknown element is the score function

$$s(x, t) = \nabla_x \log p(x, t), \tag{5}$$

where $p(x, t)$ is the solution to the forward and reverse SDEs. If we know the score function, we can use numerical SDE solvers to propagate samples backward in time.

**Forward process**

Diffusion generative networks usually define the latent space to be a standard Gaussian $\mathcal{N}(0, 1)$. We can construct the forward process Eq.(2) by simplifying the drift to be linear in $x$, i.e. $f(x, t) = x f(t)$. Now the SDE and the FPE read

$$dx = x\, f(t)\, dt + g(t)\, dW.$$
$$\frac{\partial p(x, t)}{\partial t} = -f(t)p(x, t) - x f(t)\nabla_x p(x, t) + \frac{g(t)^2}{2}\nabla_x^2 p(x, t). \tag{6}$$

In this case, we can analytically derive the solution of the FPE for given $f(t)$ and $g(t)$. We make the ansatz that the time evolution starting from an event $x_0 \sim p_{\text{data}}$ is a Gaussian

$$p(x, t|x_0) = \mathcal{N}(x|\mu(t), \sigma(t)) = \frac{1}{\sqrt{2\pi}\sigma(t)}\exp\left(-\frac{(x-\mu(t))^2}{2\sigma(t)^2}\right)$$
$$\Leftrightarrow \quad x(t|x_0) = \mu(t) + \sigma(t)\epsilon \quad \text{with} \quad \epsilon \sim \mathcal{N}(0, 1), \tag{7}$$

with time dependent mean $\mu(t)$ and standard deviation $\sigma(t)$. Using this ansatz in the FPE Eq.(6), we obtain

$$\frac{x-\mu}{\sigma^2}\frac{\partial\mu}{\partial t} + \frac{(x-\mu)^2}{\sigma^3}\frac{\partial\sigma}{\partial t} - \frac{1}{\sigma}\frac{\partial\sigma}{\partial t} = f(t)\left(x\frac{x-\mu}{\sigma^2} - 1\right) + \frac{g(t)^2}{2}\left(\frac{(x-\mu)^2}{\sigma^4} - \frac{1}{\sigma^2}\right)$$
$$= f(t)\frac{(x-\mu)^2 + \mu(x-\mu) - \sigma^2}{\sigma^2} + g(t)^2\frac{(x-\mu)^2 - \sigma^2}{2\sigma^4}. \tag{8}$$

Sorting this equation by powers of $(x-\mu)$ and comparing coefficients we find relations between $\mu(t), \sigma(t)$ and $f(t), g(t)$,

$$\frac{\partial\mu(t)}{\partial t} = f(t)\mu(t)$$
$$\frac{\partial\sigma(t)}{\partial t} = f(t)\sigma(t) + \frac{g(t)^2}{2\sigma(t)}. \tag{9}$$

The solutions of those two differential equations with initial conditions $\sigma(0) = 0$ and $\mu(0) = x_0$ are

$$\mu(t) = x_0\alpha(t)$$
$$\sigma(t) = \alpha(t)\left[\int_0^t dt'\frac{g(t')^2}{\alpha(t')^2}\right]^{1/2} \quad \text{with} \quad \alpha(t) = \exp\int_0^t dt'f(t'). \tag{10}$$

If these equations are fulfilled, the Gaussian ansatz is the solution to the FPE. This gives us a solution for general $f(t), g(t)$. However, only if the boundary conditions $\alpha(1) = 0$ and

$\sigma(1) = 1$ are fulfilled, the full unconditional density follows

$$p(x, 0) = \int dx_0 \, p(x, 0|x_0) \, p_{\text{data}}(x_0) = \int dx_0 \, \delta(x - x_0) \, p_{\text{data}}(x_0) = p_{\text{data}}(x)$$

$$p(x, 1) = \int dx_0 \, p(x, 1|x_0) \, p_{\text{data}}(x_0) = \mathcal{N}(x; 0, 1) \int dx_0 \, p_{\text{data}}(x_0) = \mathcal{N}(0, 1), \quad (11)$$

as specified in Eq.(1).

**Relation to CFM**

Equations (7) and (10) describe the mathematics behind all generative diffusion networks. Conditional flow matching (CFM) [37], a specific type of diffusion network that has seen a lot of success in particle physics [19, 38–42], can be derived from this formalism. First, we use the fact that for any diffusion SDE there exists an associated ODE that encodes the same time-dependent density $p(x, t)$. It can be derived by rewriting the FPE (3) as

$$\frac{\partial p(x, t)}{\partial t} = -\nabla_x \left[ \left( f(x, t) - \frac{1}{2} g(t)^2 \nabla_x \log p(x, t) \right) p(x, t) \right]$$

$$= -\nabla_x (v(x, t) p(x, t))$$

$$\text{with} \quad v(x, t) = f(x, t) - \frac{1}{2} g(t)^2 \nabla_x \log p(x, t). \quad (12)$$

This continuity equation corresponds to the ODE

$$dx = v(x, t) \, dt. \quad (13)$$

The deterministic (ODE) and stochastic (SDE) processes are equivalent in the sense that they have the same density solution $p(x, t)$. The difference between the SDE drift $f(x, t)$ and the ODE velocity $v(x, t)$ is that the former can be hand-crafted such that the forward SDE transports from the data to the latent distribution. To generate samples the score function $s(x, t)$ is also required. The velocity field combines the forward drift and the score function of the underlying process into one time-symmetric description. For known $f(x, t)$ and $g(t)$ the velocity and the score functions can be converted into each other.

CFMs work directly with the velocity field $v$ of the ODE, the underlying SDE is not explicitly constructed. Instead, the trajectories Eq.(7) are used to define a forward process from data to noise. Following Ref. [38], we set $\alpha(t) = (1 - t)$ and $\sigma(t) = t$, defining linear trajectories

$$x(t|x_0) = (1 - t)x_0 + t\epsilon. \quad (14)$$

We then encode the ODE-velocity in a network using an MSE loss

$$\mathcal{L}_{\text{CFM}} = \left\langle [v_\theta(x, t) - v(x, t|x_0)]^2 \right\rangle_{t \sim \mathcal{U}(0,1), x_0 \sim p_{\text{data}}(x_0), \epsilon \sim \mathcal{N}}. \quad (15)$$

For a detailed derivation of the formalism and the loss function see e.g. Refs. [23, 38]. In practice, we can use a wealth of network architectures to encode the velocity, from a simple fully connected network [38], to transformers [39], vision transformers [42], and Lorentz-equivariant transformers [43, 44].

## 2.2 Distribution to distribution

We now extend the SDE formalism to two arbitrary distributions. The generative direction starts from the initial $p(x_0)$ and samples the $p(x_1)$. The goal is to find a drift function $f(x, t)$ such that the SDE moves from $x_0 \sim p(x_0)$ at time $t = 0$ to $x_1 \sim p(x_1)$ at time $t = 1$.

**Doob's h-transform**

The key ingredient to this generalization is Doob's $h$-transform [45]. It conditions a given SDE, called the reference process, on a pre-defined final point. The reference process follows an SDE like Eq.(2),

$$dx_{\text{ref}} = f(x, t) \, dt + g(t) \, dW \, . \tag{16}$$

From $t = 0$ to $t = 1$ it encodes a time-evolving density $p_{\text{ref}}(x, t)$ for the entire stochastic process, as well as the conditional $p_{\text{ref}}(x, t|x_0)$ describing the stochastic trajectories starting from a specific $x_0$.

We modify this SDE to guarantee that the endpoint is a pre-defined $x_1$, by adding a term to the drift function,

$$dx = \left[ f(x, t) + g(t)^2 h(x, t, x_1) \right] dt + g(t) dW$$
$$\text{with} \quad h(x, t, x_1) = \nabla_x \log p_{\text{ref}}(x_1, t = 1|x) \, . \tag{17}$$

The Doob's $h$-transform function depends on the current state of the SDE $x(t)$ at time $t$ and on the specified final point $x_1$. The density $p_{\text{ref}}(x_1, t = 1|x)$ is the transition probability that the reference process reaches $x_1$ at $t = 1$ conditioned on the state $x(t)$ at time $t$. Including this term in the drift forces the trajectories to walk up the gradient of this density and pushes them towards intermediate states that are more compatible with the desired final state. This way, it corrects the initial SDE by forcing it towards the specified $x_1$.

We note that $h$ depends on the reference process through $p_{\text{ref}}(x_1, t = 1|x)$. This correction adapts to the chosen initial SDE. Different initial values $f(x, t)$ and $g(t)$ lead to a different correction from the Doob's $h$-transform, but eventually arrive in the specified $x_1$. This means we can simplify the reference process into a pure noising,

$$f(x, t) = 0 \qquad \Rightarrow \qquad dx^{\text{ref}} = g(t) \, dW \, . \tag{18}$$

For this choice we can use Eqs.(10) and (17) to calculate the $h$-transform

$$\alpha_{\text{ref}}(t) = 1$$
$$\sigma_{\text{ref}}(t)^2 = \int_t^1 dt' g(t')^2$$
$$p_{\text{ref}}(x_1, t = 1|x) = \mathcal{N}\left(x_1; x, \sigma_{\text{ref}}(t)\right) \propto \exp\left[-\frac{1}{2} \frac{(x_1 - x(t))^2}{\sigma_{\text{ref}}(t)^2}\right]$$
$$h(x, t, x_1) = \frac{x_1 - x(t)}{\sigma_{\text{ref}}(t)^2} \, . \tag{19}$$

We have explicitly constructed a forward SDE

$$dx = \frac{g(t)^2}{\sigma_{\text{ref}}(t)^2}(x(t) - x_1) \, dt + g(t) \, dW \, , \tag{20}$$

for which the solution trajectories are guaranteed to end in $x_1 \sim p_1$. According to Eq.(3) the underlying probability distribution fulfills

$$\frac{\partial p(x, t|x_1)}{\partial t} = -\nabla_x \frac{g(t)^2(x(t) - x_1)p(x, t|x_1)}{\sigma_{\text{ref}}(t)^2} + \frac{g(t)^2}{2} \nabla_x^2 p(x, t|x_1) \, . \tag{21}$$

Using the same method, we can also describe a reverse process, for which the solution trajectories end in $x_0 \sim p_0$. The reference process

$$d\bar{x}_{\text{ref}} = g(t)\, d\bar{W} \,, \tag{22}$$

moves from $t = 1$ to $t = 0$. In complete analogy, it encodes the conditional probability $\bar{p}_{\text{ref}}(\bar{x}(t)|x_1)$ starting from a specific point $x_1$. Applying the $h$-transform leads to the SDE

$$d\bar{x} = \left[-g(t)^2 \bar{h}(\bar{x}, t, x_0)\right] dt + g(t) d\bar{W}$$
$$\text{with} \quad \bar{h}(\bar{x}, t, x_0) = \nabla_{\bar{x}} \log \bar{p}_{\text{ref}}(x_0, t = 0|\bar{x}) \,, \tag{23}$$

and modifies Eq.(19) to

$$\bar{\sigma}_{\text{ref}}(t)^2 = \int_0^t dt' g(t')^2$$
$$\bar{h}(\bar{x}, t, x_0) = \frac{x_0 - \bar{x}(t)}{\bar{\sigma}_{\text{ref}}(t)^2} \,. \tag{24}$$

Again we constructed a generating SDE

$$d\bar{x} = \frac{g(t)^2}{\bar{\sigma}_{\text{ref}}(t)^2}(\bar{x}(t) - x_0)\, dt + g(t)\, d\bar{W} \,. \tag{25}$$

whose solutions end in $x_0$ and the underlying probability density follows

$$\frac{\partial \bar{p}(x, t|x_0)}{\partial t} = -\nabla_{\bar{x}} \frac{g(t)^2 (\bar{x}(t) - x_0) \bar{p}(\bar{x}, t|x_0)}{\bar{\sigma}_{\text{ref}}(t)^2} - \frac{g(t)^2}{2} \nabla_{\bar{x}}^2 \bar{p}(\bar{x}, t|x_0) \,. \tag{26}$$

So far, we have constructed the forward and the reverse processes independently. We now assume that they are the time-reversal of each other and that the forward and reverse FPEs (21) and (26) have a common Gaussian solution

$$p(x, t|x_0, x_1) = \bar{p}(\bar{x}, t|x_0, x_1) = \mathcal{N}(x|\mu(t), \sigma(t)) \,. \tag{27}$$

Inserting this ansatz into the forward FPE (21), we obtain

$$\frac{x - \mu}{\sigma^2} \frac{\partial \mu}{\partial t} + \frac{(x - \mu)^2}{\sigma^3} \frac{\partial \sigma}{\partial t} - \frac{1}{\sigma} \frac{\partial \sigma}{\partial t} = \frac{g^2}{\sigma_{\text{ref}}^2} \left( \frac{(x_1 - x)(x - \mu)}{\sigma^2} + 1 \right) + \frac{g^2}{2} \left( \frac{(x - \mu)^2}{\sigma^4} - \frac{1}{\sigma^2} \right) \,. \tag{28}$$

It is solved by

$$\frac{\partial \mu(t)}{\partial t} = \frac{g(t)^2 (x_1 - \mu(t))}{\sigma_{\text{ref}}(t)^2}$$
$$\frac{\partial \sigma(t)}{\partial t} = -\frac{g(t)^2 \sigma(t)}{\sigma_{\text{ref}}(t)^2} + \frac{g(t)^2}{2\sigma(t)} \,. \tag{29}$$

From the reverse FPE (26) we find the corresponding

$$\frac{\partial \mu(t)}{\partial t} = \frac{g(t)^2 (\mu(t) - x_0)}{\bar{\sigma}_{\text{ref}}(t)^2}$$
$$\frac{\partial \sigma(t)}{\partial t} = \frac{g(t)^2 \sigma(t)}{\bar{\sigma}_{\text{ref}}(t)^2} - \frac{g(t)^2}{2\sigma(t)} \,. \tag{30}$$

Equating Eq.(29) and Eq.(30) yields

$$\mu(t) = \frac{\bar{\sigma}_{\text{ref}}(t)^2 x_1 + \sigma_{\text{ref}}(t)^2 x_0}{\bar{\sigma}_{\text{ref}}(t)^2 + \sigma_{\text{ref}}(t)^2}$$

$$\sigma(t) = \sqrt{\frac{\bar{\sigma}_{\text{ref}}(t)^2 \sigma_{\text{ref}}(t)^2}{\bar{\sigma}_{\text{ref}}(t)^2 + \sigma_{\text{ref}}(t)^2}} \, . \tag{31}$$

This solves Eq.(29) and Eq.(30) with the boundary conditions $\sigma(0) = \sigma(1) = 0$, $\mu(0) = x_0$ and $\mu(1) = x_1$. Finally, the conditional probability of both processes is given by

$$p(x(t), t | x_0, x_1) = \mathcal{N}\left(x \middle| \frac{\bar{\sigma}_{\text{ref}}(t)^2 x_1 + \sigma_{\text{ref}}(t)^2 x_0}{\bar{\sigma}_{\text{ref}}(t)^2 + \sigma_{\text{ref}}(t)^2}, \sqrt{\frac{\bar{\sigma}_{\text{ref}}(t)^2 \sigma_{\text{ref}}(t)^2}{\bar{\sigma}_{\text{ref}}(t)^2 + \sigma_{\text{ref}}(t)^2}}\right)$$

$$\propto \mathcal{N}(x | x_0, \bar{\sigma}_{\text{ref}}) \, \mathcal{N}(x | x_1, \sigma_{\text{ref}}) \, . \tag{32}$$

**Loss function**

The full unconditional density encoded in the constructed stochastic process is obtained by marginalizing over the conditions.

$$p(x, t) = \int dx_0 \, dx_1 \, p^{\text{train}}(x_0, x_1) \, p(x, t | x_0, x_1)$$

$$= \int dx_0 \, dx_1 \, p^{\text{train}}(x_0, x_1) \, \mathcal{N}\left(x | \mu(t), \sigma(t)\right) = \begin{cases} p_0(x) & t = 0 \\ p_1(x) & t = 1 \, . \end{cases} \tag{33}$$

The joint distribution $p^{\text{train}}(x_0, x_1)$ is defined by the pairing in the training data, in the case of unpaired data it factorizes to $p^{\text{train}}(x_0, x_1) = p_0(x_0)p_1(x_1)$. Both limits of the stochastic process are fulfilled independent of the choice of joint distribution. For instance, for unfolding, we can use the pairing between reco-level and gen-level events from the forward simulation.

To construct a generative network, we need to remove the conditions on the two end points. This means we want to find an SDE that encodes the distribution from Eq.(33), but with a drift function that only depends on the current state of the SDE. We can derive this unconditional drift term similarly to the unconditional CFM-velocity [37, 38], starting with the FPE (3),

$$\frac{\partial p(x, t)}{\partial t} = \int dx_0 dx_1 p^{\text{train}}(x_0, x_1) \frac{\partial p(x, t | x_0, x_1)}{\partial t}$$

$$= \int dx_0 dx_1 p^{\text{train}}(x_0, x_1) \left[ -\nabla_x [f(x, t | x_0, x_1) p(x, t | x_0, x_1)] + \frac{g^2}{2} \nabla_x^2 p(x, t | x_0, x_1) \right]$$

$$= -\nabla_x \left[ p(x, t) \int dx_0 dx_1 p^{\text{train}}(x_0, x_1) \frac{f(x, t | x_0, x_1) p(x, t | x_0, x_1)}{p(x, t)} \right]$$

$$+ \frac{g^2}{2} \nabla_x^2 \int dx_0 dx_1 p^{\text{train}}(x_0, x_1) p(x, t | x_0, x_1)$$

$$= -\nabla_x [p(x, t) f(x, t)] + \frac{g^2}{2} \nabla_x^2 p(x, t) \, , \tag{34}$$

where we define

$$f(x, t) = \int dx_0 dx_1 p^{\text{train}}(x_0, x_1) \frac{f(x, t | x_0, x_1) p(x, t | x_0, x_1)}{p(x, t)} \, . \tag{35}$$

With this drift function and a diffusion term $g(t)$, the solution of the FPE is given by Eq.(33). This gives us an SDE which propagates samples between $x_1 \sim p_1$ and $x_0 \sim p_0$, only depending on the current state $x(t)$. Starting from one of the distributions and numerically solving the SDE generates samples from the other distribution.

The last problem with the drift in Eq.(35) is that we cannot evaluate it analytically, so we encode it into a network $f_\theta$. For this regression problem the natural loss is the MSE, but this requires training samples $f(x, t)$. We re-write this objective in terms of the conditional drift $f(x, t | x_0, x_1)$ defined in the SDE Eq.(20) and the conditional trajectories $p(x, t | x_0, x_1)$ defined in Eq.(39), as these allow for efficient generation of training samples. Following all steps from Ref. [38] the distribution mapping loss becomes

$$
\mathcal{L}_{\text{DM}} = \left\langle \left( f_\theta(x, t) - f(x, t | x_0, x_1) \right)^2 \right\rangle_{t,(x_0,x_1) \sim p^{\text{train}}(x_0,x_1), x \sim p(x,t|x_0,x_1)}
$$
$$
= \left\langle \left( f_\theta(x, t) - \frac{g(t)^2 (x - x_0)}{\bar{\sigma}(t)^2} \right)^2 \right\rangle_{t,(x_0,x_1) \sim p^{\text{train}}(x_0,x_1), x \sim p(x,t|x_0,x_1)} . \tag{36}
$$

The learned drift function depends on the pairing information in the training data, encoded via $p^{\text{train}}(x_0, x_1)$. Different pairings lead to different SDEs encoding different trajectories, but they all result in a generative network with the correct boundary distributions in Eq.(33).

**Noise schedules for SB and DiDi**

Choosing $g(t) = \sqrt{\beta(t)}$, with $\beta(t)$ the triangular function

$$
\beta(t) = \begin{cases} \beta_0 + 2(\beta_1 - \beta_0)t & 0 \le t < \dfrac{1}{2} \\ \beta_1 - 2(\beta_1 - \beta_0)\left(t - \dfrac{1}{2}\right) & \dfrac{1}{2} \le t \le 1 \end{cases} \tag{37}
$$

and $\beta_0 = 10^{-5}$ and $\beta_1 = 10^{-4}$, we obtain the SB formulation [18, 46].

For constant $g(t) = g$, Eq.(19) simplifies to $\bar{\sigma}_{\text{ref}}(t) = g\sqrt{t}$ and $\sigma_{\text{ref}}(t) = g\sqrt{1-t}$. Consequently, Eq.(31) yields

$$
\mu(t) = (1 - t)x_0 + t x_1 \qquad \text{and} \qquad \sigma(t) = g\sqrt{t(1-t)} . \tag{38}
$$

Our trajectory and probability distribution take the form

$$
x(t) = (1 - t)x_0 + t x_1 + g\sqrt{t(1-t)}\, \epsilon \qquad \text{with} \ \epsilon \sim \mathcal{N}(0, 1)
$$
$$
\Leftrightarrow \qquad p(x(t), t | x_0, x_1) = \mathcal{N}\left( x(t) | (1 - t)x_0 + t x_1, g\sqrt{t(1-t)} \right) . \tag{39}
$$

The noise term vanishes at the endpoints $t = 0, 1$ and takes its maximum at $t = 1/2$. This constructs an SDE that interpolates between two arbitrary distributions and reduces to DiDi [19] for $g \to 0$. To see this we start from Eq.(20) and insert the training trajectory from Eq.(39),

$$
dx(t) = \frac{x(t) - x_0}{t}\, dt + g\, dW
$$
$$
= \frac{(1 - t)x_0 + t x_1 + g\sqrt{t(1-t)}\, \epsilon - x_0}{t}\, dt + g\, dW
$$
$$
= (x_1 - x_0)\, dt + g\left[ \frac{\sqrt{t(1-t)}\, \epsilon}{t}\, dt + dW \right] . \tag{40}
$$

For $g \to 0$ the training SDE reduces to the training ODE from DiDi, with the linear velocity field $x_1 - x_0$ and the distribution mapping loss reduces to the flow matching loss.

## 2.3 Conditional distribution mapping

In the last section we have constructed an SDE-based mapping between two arbitrary distributions. We now describe the new aspect of adapting this DM-formalism to reproduces the correct conditional distributions [34]. Specifically, we look at the trajectories $p(x, t|x_1)$ obtained when solving the learned SDE repeatedly from the same starting point $x_1 \sim p_1$, and modify our formalism such that for $t \to 0$ it reproduces the correct data pairing $p^{\text{train}}(x_0|x_1)$.

First, we check how this density looks in our conditional training trajectories $p(x(t), t|x_0, x_1)$ by marginalizing over $x_0$. Similar to Eq.(33) we can write

$$p(x, t|x_1) = \int dx_0 \, p^{\text{train}}(x_0|x_1) \, p(x(t), t|x_0, x_1)$$

$$= \int dx_0 \, p^{\text{train}}(x_0|x_1) \, \mathcal{N}\big(x|\mu(t), \sigma(t)\big) = \begin{cases} p^{\text{train}}(x|x_1) & t = 0 \\ \delta(x_1 - x) & t = 1 \,. \end{cases} \quad (41)$$

This conditional density has the correct boundary behavior. When conditioned on a latent-space event $x_1 \sim p_{\text{latent}}$, the density converges to a delta peak around this event at time $t = 1$ and converges to the training pairing conditional distribution $p^{\text{train}}(x|x_1)$ at time $t = 0$.

However, there is no guarantee that the generative SDE defined via the drift from Eq.(35) shares these conditional densities. We have derived this drift $f$ by showing that it solves the same unconditional FPE in Eq.(34) as our training process, so we are only sure that they share the unconditional density $p(x, t)$.

We need a drift function leading to an SDE that shares the conditional density $p(x, t|x_1)$ of the training trajectories. This can be achieved by going through the same derivation as we did for the drift initially, but this time with the FPE for the conditional density

$$\frac{\partial p(x, t|x_1)}{\partial t} = \int dx_0 p^{\text{train}}(x_0|x_1) \frac{\partial p(x, t|x_0, x_1)}{\partial t}$$

$$= \int dx_0 p^{\text{train}}(x_0|x_1) \left[ -\nabla_x [f(x, t|x_0, x_1) p(x, t|x_0, x_1)] + \frac{g^2}{2} \nabla_x^2 p(x, t|x_0, x_1) \right]$$

$$= -\nabla_x \left[ p(x, t|x_1) \int dx_0 p^{\text{train}}(x_0|x_1) \frac{f(x, t|x_0, x_1) p(x, t|x_0, x_1)}{p(x, t|x_1)} \right]$$

$$+ \frac{g^2}{2} \nabla_x^2 \int dx_0 p^{\text{train}}(x_0|x_1) p(x, t|x_0, x_1)$$

$$= -\nabla_x [p(x, t|x_1) f(x, t|x_1)] + \frac{g^2}{2} \nabla_x^2 p(x, t|x_1) \,. \quad (42)$$

Here, we identify the conditional drift term to be

$$f(x, t|x_1) = \int dx_0 \, p^{\text{train}}(x_0|x_1) \frac{f(x, t|x_0, x_1) p(x, t|x_0, x_1)}{p(x, t|x_1)} \,. \quad (43)$$

Sampling a latent $x_1 \sim p_{\text{latent}}$ and solving the SDE with this drift function samples from the conditional density $p^{\text{train}}(x|x_1)$. This new conditional drift is not only a function of the current state of the SDE, but also of the initial state $x_1$. This additional input acts as the condition under which we want to generate a sample from the data distribution, similar to the way the CFM velocity is a function of the current state of the ODE and of the condition.

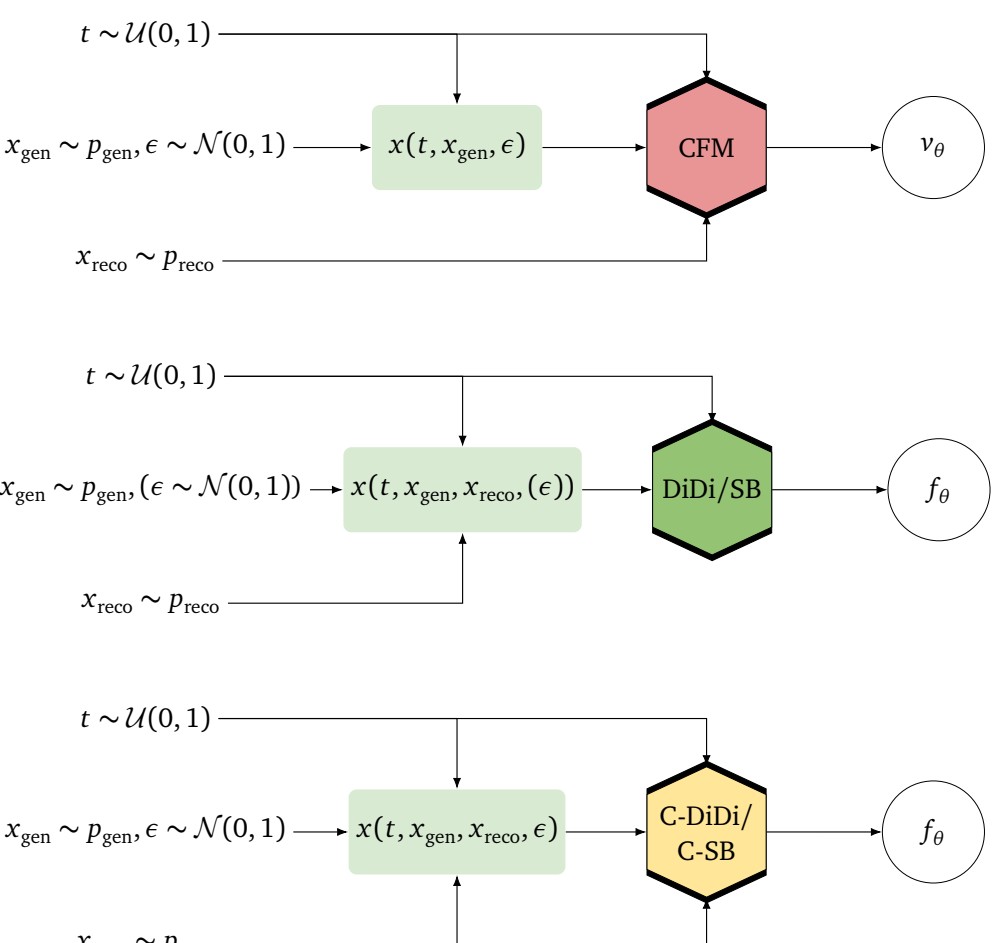

Figure 1: Schematic illustration of the training procedure of a CFM-based (top), a distribution mapping-based (middle) and a conditional distribution mapping-based (bottom) generative unfolding pipeline.

**Loss function**

This conditional drift is once again encoded in a network. Repeating the derivation of Ref. [38], we obtain the conditional distribution mapping (CDM) loss

$$
\begin{aligned}
\mathcal{L}_{\mathrm{CDM}} &= \left\langle \left( f_\theta(x, t, x_1) - f(x, t | x_0, x_1) \right)^2 \right\rangle_{t,(x_0,x_1)\sim p(x_0,x_1), x\sim p(x,t|x_0,x_1)} \\
&= \left\langle \left( f_\theta(x, t, x_1) - \frac{g(t)^2(x - x_0)}{\bar\sigma(t)} \right)^2 \right\rangle_{t,(x_0,x_1)\sim p^{\mathrm{train}}(x_0,x_1), x\sim p(x,t|x_0,x_1)} .
\end{aligned}
\tag{44}
$$

It is identical to the standard DM-loss, except for the network also using the initial condition $x_1$ as a third input. This is the only change necessary to allow the network to learn the proper conditional densities.

## 2.4 Unfolding

For unfolding, we want to transform a measured reco-level distribution $p_{\mathrm{reco}}$ to the corresponding gen-level distribution $p_{\mathrm{gen}}$. Recent implementations [23] of generative unfolding use a CFM network to generate samples from the posterior distribution $p(x_{\mathrm{gen}}|x_{\mathrm{reco}})$. It learns the velocity $v(x, t, x_{\mathrm{reco}})$, linked to the posterior distribution via Eq.(12) and flowing between

a point of a Gaussian latent space and a point in the gen-level phase space conditioned on a given rec-level event. A schematic illustration of the training procedure is shown in the top part of Fig. 1.

Alternatively, we can also map reco-level events directly to their gen-level counterpart either using the SB or DiDi. The key difference to standard generative unfolding is that we use the reco-level information to define the trajectories instead of treating it as an additional input to the network. This is visualized in the center of Fig. 1. We learn the drift term of the probability $p(x, t)$ as described in Eq.(36). Optionally, we can add Gaussian noise to the trajectories to make the networks stochastic rather than deterministic. The impact of the noise is governed by the choice of the diffusion term $g(t)$.

Finally, we can combine the conditional generative approach with the DM by giving the reco-level information to the network directly. The training objective is to learn a drift term linked to the conditional probability $p(x(t), t|x_{\text{reco}})$ as in Eq.(44). In this scenario adding noise is not optional. The exact training procedure is illustrated in the lower part of Fig. 1.

## 3 Gaussian Example

To illustrate the motivation for using conditional DM for unfolding, we turn to a simple example: a Gaussian mixture model ('double Gaussian') with equally weighted components, unit variances, and means at $\mu_1 = -4$ and $\mu_2 = 4$. We run this distribution through an extreme hypothetical detector which inverts all observations

$$f(x) = -x . \tag{45}$$

Since our initial distribution is symmetric around $x = 0$, this detector leaves the marginal distribution unchanged, but allows us to see how the mapping is learned by the network.

We compare the mapping constructed by the standard and conditional Schrödinger Bridge (SB). The results are shown in Fig. 2. Each of the displayed plots is divided into three panels.

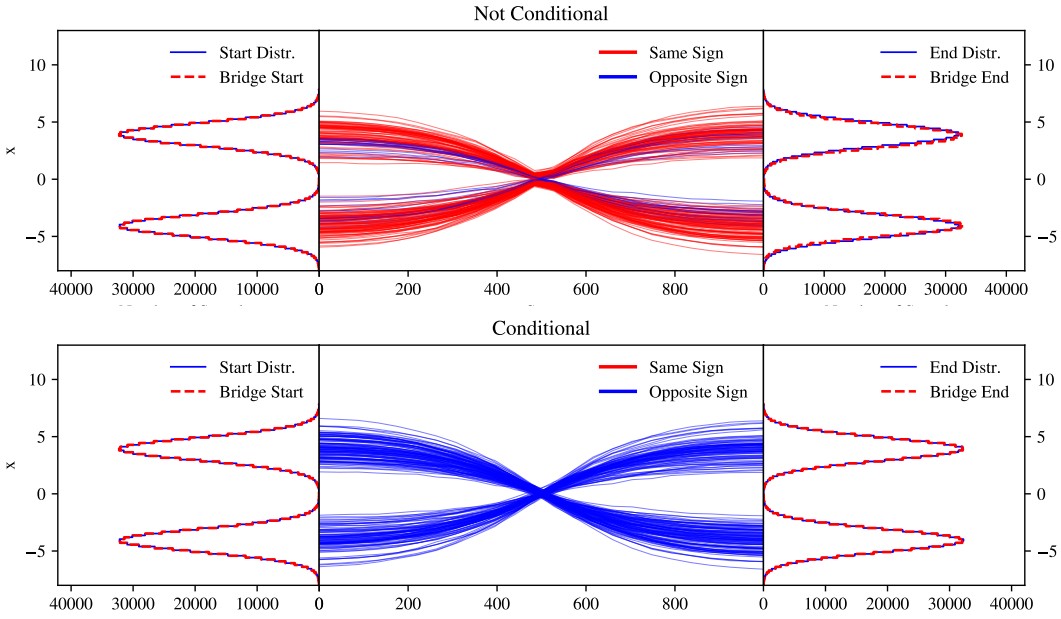

Figure 2: Non-conditional (top) vs conditional (bottom) distribution mapping when learning a $x \rightarrow -x$ mapping of a double-Gaussian.

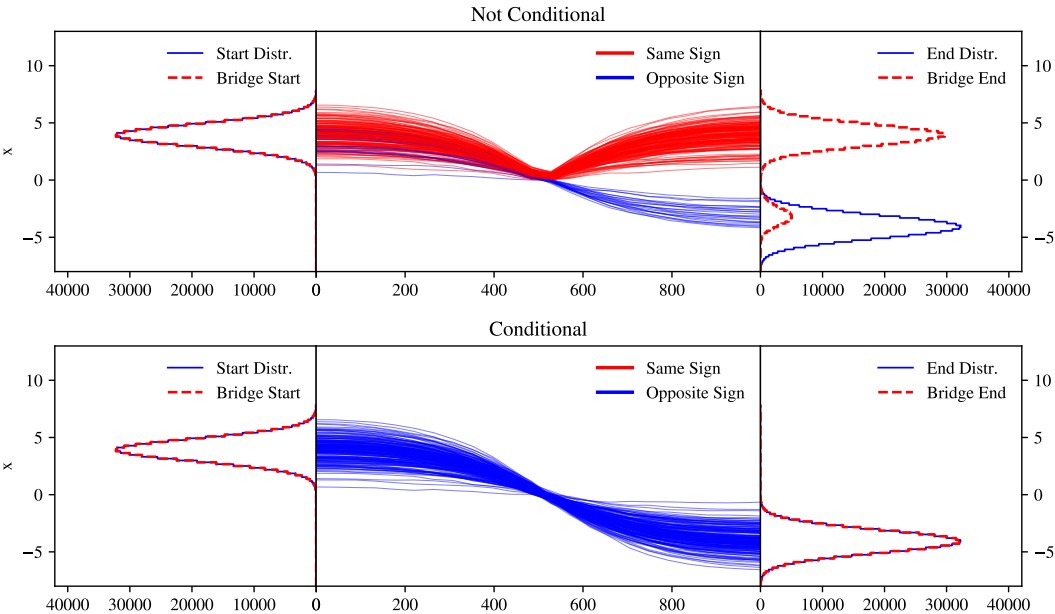

Figure 3: Non-conditional (top) vs conditional (bottom) distribution mapping when learning a $x \rightarrow -x$ mapping of a double-Gaussian, applied only to the positive peak

The left panels show the marginal distributions for the staring double Gaussian (blue, solid) and the starting point of the SB network (red, dotted). The right panels show the marginals of the double Gaussian after the inversion function is applied (blue, solid), as well as of the final step of the SB (red, dotted). The central panels show the SB trajectories which transform a subset of points. Each path is color coded to indicate whether the sign of the transported point changes; a blue line corresponds to a correct mapping $f(x) = -x$, while a red line indicates an incorrect mapping $f(x) \sim x$.

In the upper part of Fig. 2, we see that for the unconditional SB a large number of mappings are incorrect. Notably, the mappings can be seen to converge around $x = 0$ after half the number of total steps, and diverge after. However, the majority of paths do not cross over from $x$ to $-x$, as required, but return to the original peak. This is because at $x = 0$, both paths intersect. Since the network moves points one step at a time, it no longer has any information where to map a point once it reaches $x = 0$. As discussed in Sec. 2.3, we can break this degeneracy by providing the original starting point as input. The lower part of Fig. 3 shows the result of a conditional SB network. We see that all paths are blue, indicating a correct mapping.

So far, the marginal distribution is not impacted by the mis-modeled mapping. The network still reproduces the overall target distribution available during training. However, the mapping can present an issue to the quality of the final marginal distribution if the data the network is unfolding differs in composition to the training data. We illustrate an extreme case in Fig. 3. In the top part we apply the unconditional bridge, trained on the full double Gaussian, only to the positive peak. Now, the unconditional SB produces a wrong marginal distribution, where only a small fraction of points is mapped to the correct negative peak, and the majority remains in the positive peak. In contrast, the lower part of Fig. 3 shows that the conditional SM learns the correct mapping and therefore easily produces the correct marginal distribution.

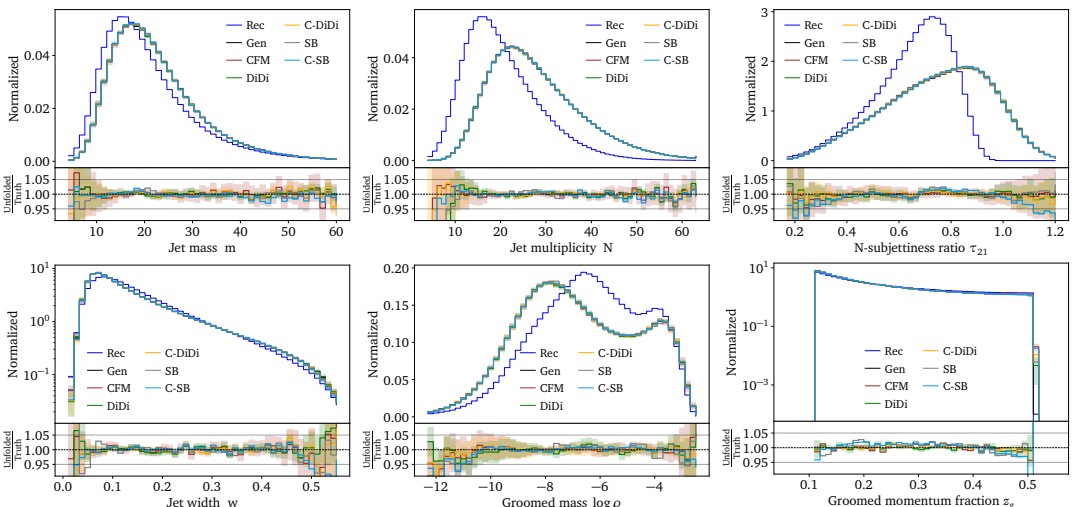

Figure 4: Unfolded distributions of the 6d jet-substructure dataset using CFM, DiDi, C-DiDi, SB and C-SB. All unfolded distributions reproduce the truth at percent level. The remaining differences are well covered by the BNN uncertainties.

# 4 Unfolding Jet Substructure Observables

As a first physics example, we consider the updated version [47] of the OmniFold dataset [8] which has become a standard benchmark for unfolding methods [11, 18, 23]. It consists of events describing

$$pp \rightarrow Z + \text{jets} \tag{46}$$

production at $\sqrt{s} = 14$ TeV. The events are generated and decayed with Pythia 8.244 [48] with Tune 26, the detector response is simulated with Delphes 3.5.0 [49] with the CMS card, that uses particle flow reconstruction. At both pre-detector (gen level) and post-detector (reco level) jets are clustered with the anti-$k_T$ algorithm [50] with $R = 0.4$, as implemented in FastJet 3.3.2 [51].

We unfold six jet substructure observables of the leading jet: mass $m$, width $\tau_1^{(\beta=1)}$, multiplicity $N$, soft drop mass [52, 53] $\rho = m_{\text{SD}}^2/p_T^2$, momentum fraction $z_g$ using $z_{\text{cut}} = 0.1$ and $\beta = 0$, and the $N$-subjettiness ratio $\tau_{21} = \tau_2^{(\beta=1)}/\tau_1^{(\beta=1)}$ [54]. The dataset contains about 24M simulated events, 20M for training and 4M for testing.

## 4.1 Unfolded distributions

We unfold the 6-dimensional phase space using all five network implementations of the three methods

- conditional generative (Conditional Flow Matching, CFM);
- unconditional distribution mapping DiDi and SB; and
- conditional distribution mapping C-DiDi and C-SB.

The respective velocity fields and drift terms are encoded in standard MLPs, the hyperparameters are given in Tab. 1. All networks are implemented in PyTorch [55] and trained with the Adam [56] optimizer. We follow the preprocessing from Ref. [23].

Due to the varying numerical requirements of different networks, we choose suitable numerical solvers for their evaluation. For the CFM, an ODE-based network, we unfold reco-level

samples using a numerical ODE-solver [57]. The SDE-based networks C-DiDi and C-SB use the DDPM SDE-solver [58]. For the unconditional DiDi and SB we have the choice between an ODE-based formulation with noise scale $g = 0$ and an SDE-based formulation with $g > 0$. We observe no significant difference in performance between them, the shown results use the SDE formulation.

In Fig. 4 we show all unfolded distribution together with the true gen-level and reco-level distributions. For CFM, DiDi and SB, we reproduce the results shown in Ref. [23]. Both methods can reliably solve this unfolding task to sub-percent precision. The new C-DiDi and C-SB give a precision on par with the established CFM method. For all networks, we only observe significant deviations from the truth far into the tails or at hard edges, for instance, in the groomed momentum fraction $z_g$.

The uncertainties reported in the figures are produced from posterior sampling. This is possible by making all of the models Bayesian neural networks, approximated with independent Gaussians for every network parameter, doubling the number of model parameters [59–63]. Previous studies in the context of the LHC have shown that the posterior is a reasonable estimate of the variation introduced by the limited size of the training dataset [64–66]. Even though the weights are Gaussian distributed, the final network output is generally not a Gaussian. The concept of BNNs can also be applied to the density estimation in generative networks [38, 67, 68], including diffusion generators [19, 38].

The uncertainties shown in Fig. 4 are obtained by evaluating the respective networks 20 times, each time with a new set of network weights sampled from the learned distribution. The deviations from the true gen-level distribution as well as the differences between the five networks are generally covered by these uncertainties. As expected, the uncertainty increases in regions of low training statistics e.g. the tail of the jet mass distribution $m$.

## 4.2 Learned Mapping

Next, we check if the learned mapping between the reco-level and gen-level distributions agrees with the physical forward simulation. We show migration matrices for some of the observables in Fig. 5: the first row shows the truth encoded in the training data; the following columns show the learned event-wise mapping from the the CFM, unconditional DiDi/SB and C-DiDi/C-SB. While the 6-dimensional unfolded distributions are nearly identical for all methods, the migration plots show a significant difference between the unconditional and conditional networks.

The conditional CFM generator is, by design, trained to reproduce the conditional distribution $p(x_\text{gen}|x_\text{reco})$. In contrast, the unconditional DM learns to map $p(x_\text{reco}) \to p(x_\text{gen})$, but with an unphysically diagonal optimal transport prescription, as showcased in the third row. Finally, we see that the conditional C-DiDi and C-SB encode the conditional probabilities just like the CFM does.

Finally, we take a closer look at the learned posterior distributions $p(x_\text{gen}|x_\text{reco})$. While single-event-unfolding is an ill-defined analysis task, we can use the per-event posterior to illustrate the performance of the different unfolding generators [39, 69]. All our methods are inherently non-deterministic when inputting the same reco-level event repeatedly, which allows us to generate non-trivial learned posteriors. For the CFM, we sample the latent Gaussian distribution, for any one given latent space point the ODE trajectory is deterministic. In contrast, the DM-methods always start from the same latent space point, the reco event, but evolve it using a non-deterministic SDE.

In Fig. 6, we show some single-event posterior distributions from the three methods, obtained by unfolding the same reco-event 10000 times. For reference, we include the unfolded

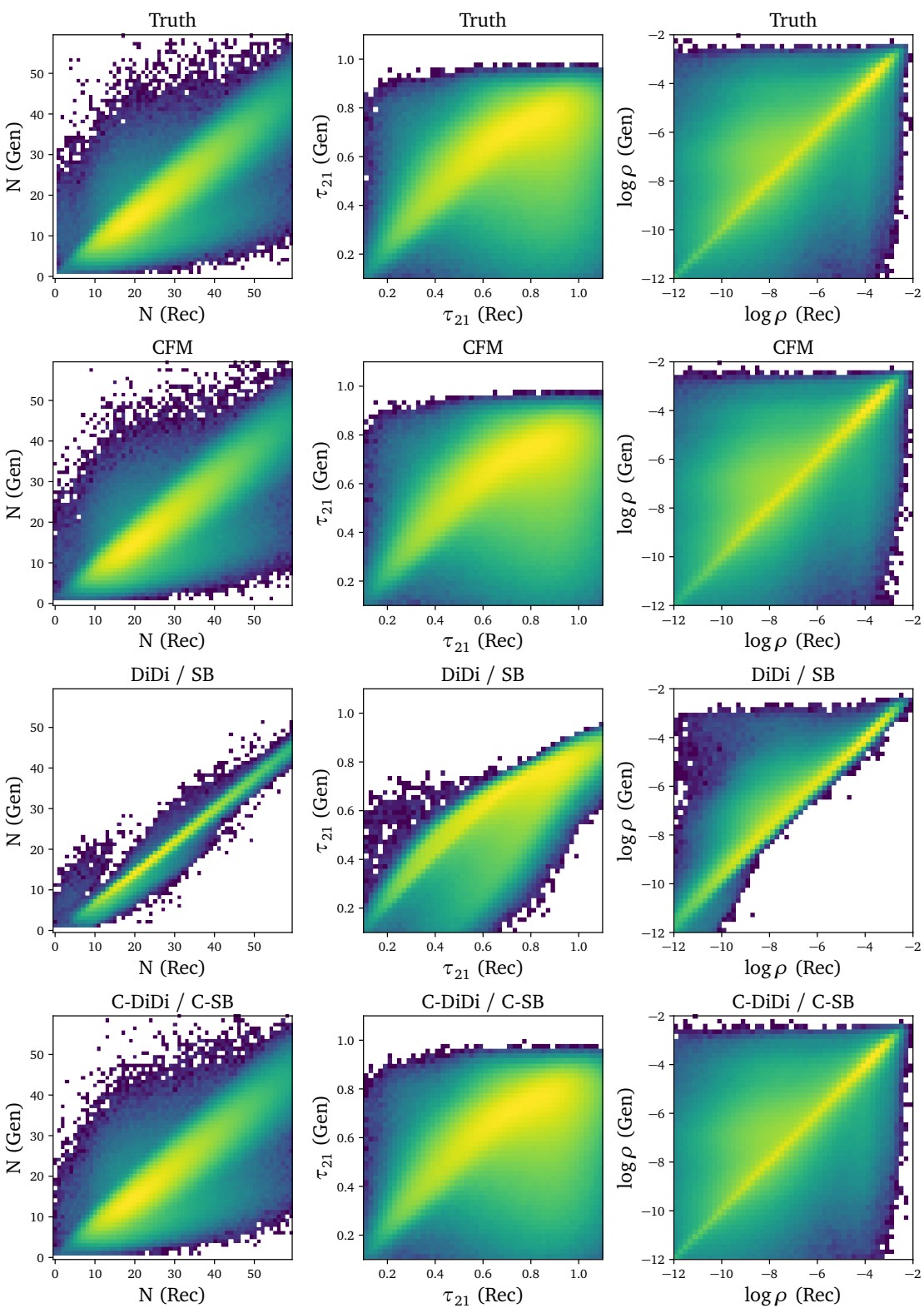

Figure 5: Migration maps for the 6D OmniFold dataset, truth compared to three different methods. We only show DiDi and C-DiDi, after verifying that the results are indistinguishable from SB and C-SB.

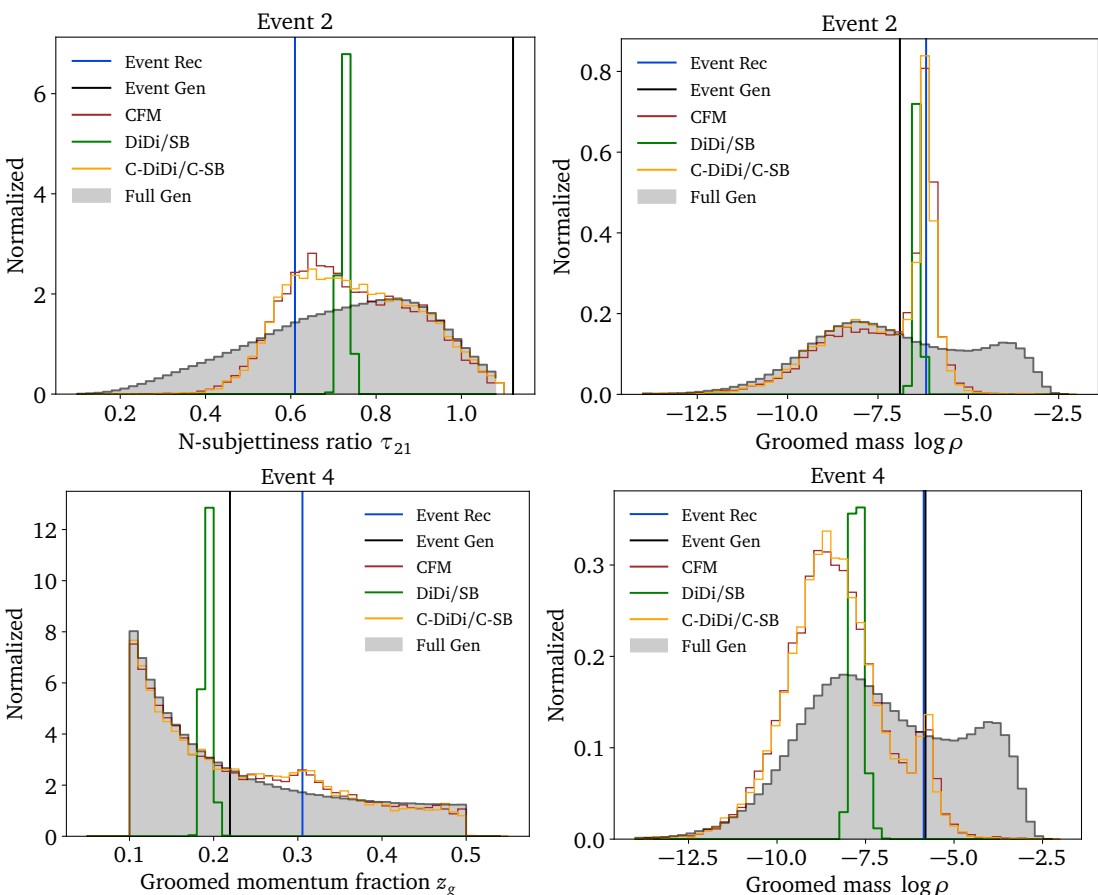

Figure 6: Posterior distributions obtained from unfolding single events with CFM, DiDi and C-DiDi on the 6D OmniFold dataset. Each of the three events is unfolded 10000 times. For reference, we also show the full gen-level distribution.

event at reco-level, the gen-level truth, and the full unconditional gen-level distribution. Again, we observe a different behavior between the unconditional DM on the one hand and the CFM and conditional DM on the other. As expected from the derivation and from the migration plots, the unconditional DiDi network does not learn a physics-defined posterior, and our sampled distribution shows simply Gaussian smearing. The width of the Gaussian is related to the noise scale $g$ of the SDE, which we verified by varying the noise scale over four orders of magnitude. The two conditional methods learn physically meaningful posteriors. Their shapes vary widely for the shown events and observables, but they agree between the different methods. We checked that the C-DiDi posteriors are invariant when varying the SDE noise scale $g$, so the learned single-event unfoldings illustrate how the CFM and the conditional DM-methods learn the same non-trivial conditional posteriors.

## 4.3 Classifier test

One approach to quantitatively test the performance of unfolding across the entire measured phase space is to use a post-hoc classifier, assuming that supervised classifier training is more effective than unsupervised density estimation[†] as part of the generative networks [23, 71]. A well-trained and calibrated classifier $C$ comparing training and generated events will approx-

---

[†]In practice, if this is the case, the post-hoc classifier could be used to improve the quality of the generative model [70].

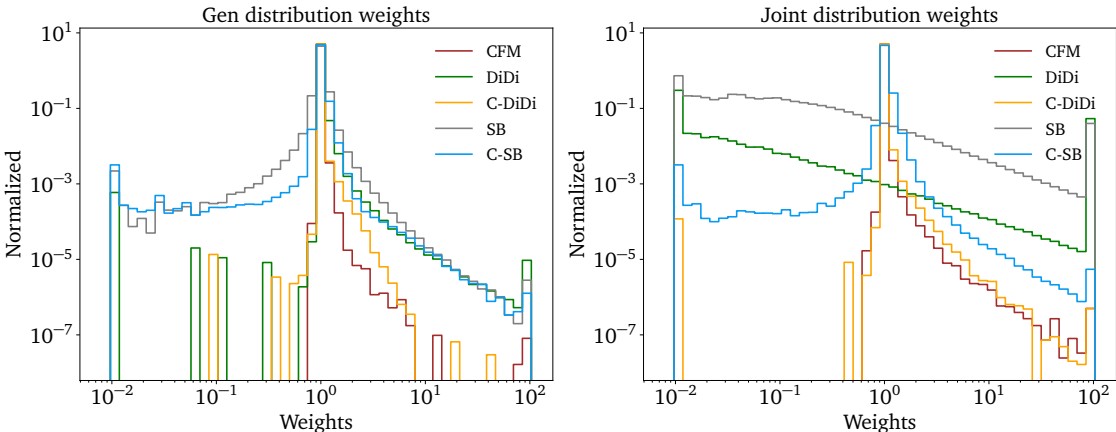

Figure 7: Classifier weight distributions for each network applied to the 6D OmniFold dataset. The left panel shows the gen-level weights according to Eq.(47), the right panel the joint distribution weights defined in Eq.(48).

imate the likelihood ratio

$$w(x_{\text{gen}}) = \frac{p_{\text{true}}(x_{\text{gen}})}{p_{\text{model}}(x_{\text{gen}})} = \frac{C(x_{\text{gen}})}{1 - C(x_{\text{gen}})} \ . \tag{47}$$

With a slight modification, we can employ this technique to evaluate the quality of our learned posterior distributions. Instead of training the classifier only on gen-level, we train on the joint reco-level and gen-level data. This gives us access to the likelihood ratio of the joint distributions. Making use of the fact that the reco-level distribution is the same for generated and true, we can write

$$\begin{aligned}
\frac{C(x_{\text{gen}}, x_{\text{rec}})}{1 - C(x_{\text{gen}}, x_{\text{rec}})} &= \frac{p_{\text{true}}(x_{\text{gen}}, x_{\text{rec}})}{p_{\text{model}}(x_{\text{gen}}, x_{\text{rec}})} \\
&= \frac{p_{\text{true}}(x_{\text{rec}})p_{\text{true}}(x_{\text{gen}}|x_{\text{rec}})}{p_{\text{model}}(x_{\text{rec}})p_{\text{model}}(x_{\text{gen}}|x_{\text{rec}})} \\
&= \frac{p_{\text{true}}(x_{\text{gen}}|x_{\text{rec}})}{p_{\text{model}}(x_{\text{gen}}|x_{\text{rec}})} \equiv w(x_{\text{gen}}|x_{\text{rec}}) \ . \tag{48}
\end{aligned}$$

Therefore, a classifier trained on the joint distributions gives us access to the likelihood ratio between the true and learned posterior distributions.

For each of our five networks we train a classifier, using the hyperparameters listed in Tab. 1. First we only look at the gen-level unfolded distributions and discriminate them from the true gen-level distribution. We show the corresponding weight distributions, evaluated on the generated events, in the left panel of Fig. 7. For all networks we see a dominant peak in one, indicating that for the overwhelming majority of events, the classifier cannot tell truth from generated. The tail towards lower weights indicates events which should not be there and which the classifier weight tries to remove, i.e. phase space regions that the network overpopulates. The right tail marks events and phase space regions underpopulated by the generative unfolding network.

Comparing the five networks and the three underlying methods, the CFM shows the smallest tails in both directions. The conditional DiDi network is almost on par with the CFM, the difference is covered by the classifier training fluctuations. Unconditional DiDi leads to a larger tail towards large weights, but hardly any events with small weights, indicating a bias in learning features or their correlations. The SB networks show a slightly larger spread in the weight

distribution, and the conditional SB is again significantly narrower than the unconditional version.

The right panel of Fig. 7 shows the weight distributions for the conditional phase space distributions. While for the conditional CFM and the conditional DM-networks the difference to the left panel is marginal, we now see that the unconditional DM-networks show little structure and large overflow bins, indicating that the learned joint distributions do not reproduce the training data.

## 5 Unfolding Substructure and Kinematic Properties

In order to stress-test the methods with a mixture of jet substructure and kinematic information, we simulated a dataset similar to the one used by a recent ATLAS analysis [32]. The resulting dataset has 22 instead of 6 dimensions, as in the previous dataset[‡].

### 5.1 New $Z + 2$ jets dataset

We now consider the process

$$pp \to Z_{\mu\mu} + 2 \text{ jets} . \tag{49}$$

We generate the events with Madgraph 5 [72], showering and hadronization are simulated with Pythia 8.311 [73], and detector effects are included via Delphes 3.5.0 [49] using the default CMS card. Jets are clustered at gen-level and reco-level using an anti-$k_T$ algorithm with $R = 0.4$ implemented in FastJet 3.3.4 [51].

We apply a set of selections resembling the ATLAS analysis: events are required to have exactly two muons with opposite charge and

$$p_{T,\mu} > 25 \text{ GeV} \qquad \text{and} \qquad m_{\mu\mu} \in [81, 101] \text{ GeV} . \tag{50}$$

Furthermore, we require at least two jets with

$$p_{T,j} > 10 \text{ GeV} \qquad \text{and} \qquad \Delta R_{\mu j} > 0.4 . \tag{51}$$

All events must pass all selections on gen and reco level (acceptance effects are small and ignored). The training set consists of 1.5M events, and the test set consists of 400k events.

Instead of restricting ourselves to jet substructure observables of the leading jet, we now unfold both kinematic information of the muons and leading jets as well as the substructure of the leading two jets. At the subjet level, we include the number of jet constituents $N$ and the subjettiness variables $\tau_1$, $\tau_2$ and $\tau_3$ [54] for each jets. In total this defines a 22-dimensional phase space to unfold into:

$$\left( (p_T, \eta, \phi)_{\mu_1, \mu_2}, (p_T, \eta, \phi, m, N, \tau_1, \tau_2, \tau_3)_{j_1, j_2} \right) . \tag{52}$$

The challenge is to reproduce all correlations, most notably the di-muon kinematics and the angular separation $R_{jj}$. In contrast to classifier-based unfolding, generative unfolding does not easily allow to over-constrain the physical phase space with redundant degrees of freedom. Instead, we choose a phase space parametrization that makes key correlations directly accessible

---

[‡]The OmniFold dataset does have the full set of jet constituents, but this is a high-dimensional variable-length set, which is beyond the scope of this paper.

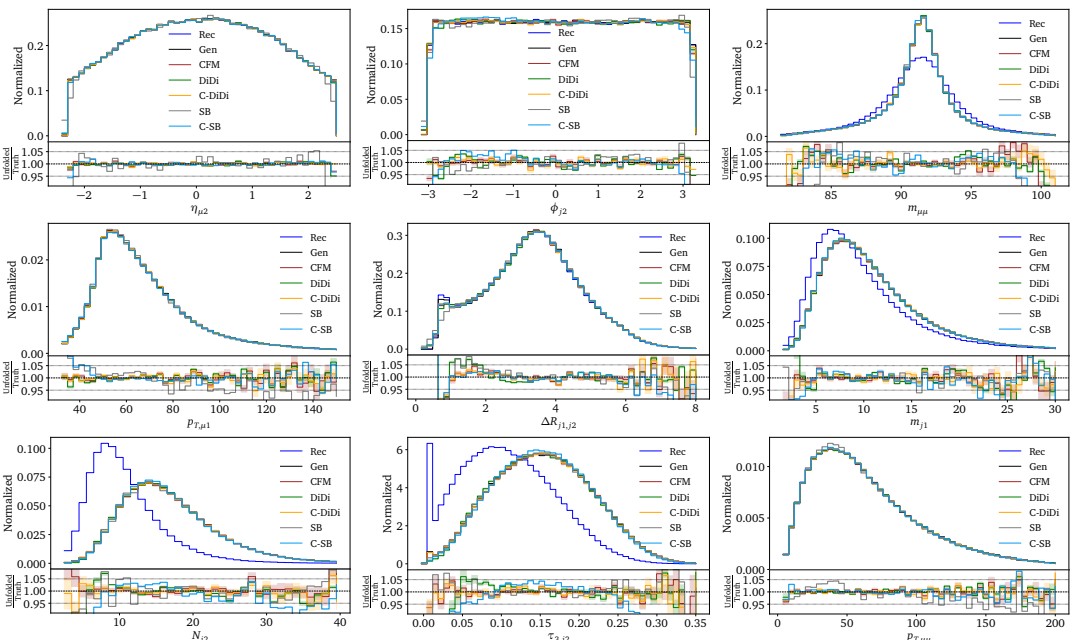

Figure 8: Unfolded distributions of the 22-dimensional Z+2 jets dataset using CFM, DiDi, C-DiDi, SB and C-SB.

to the network [20, 23]. To this end, we replace $p_{T,\mu_1} \to m_{\mu\mu}$ in the event representation and extract it later via

$$p_{T,\mu1} = \frac{m_{\mu\mu}^2}{2p_{T,\mu_2}(\cosh \Delta\eta_{\mu\mu} - \cos \Delta\phi_{\mu\mu})} \, . \tag{53}$$

We standardize our training data in each dimension. For $m_{\mu\mu}$ we apply a Breit-Wigner mapping [23].

To accommodate the more challenging phase space, we replace the MLPs in all our generators with transformers [23, 39, 43]. We follow the transformer architecture proposed in Ref. [23], the network and training hyperparameters are listed in Tab. 2 in the Appendix.

## 5.2 Unfolded distributions

The unfolding results obtained with all five networks representing all three methods are shown in Fig. 8. All of them unfold the bulk of the phase space distributions with a precision that is at the per-cent level. Small deviations from the true gen-level distributions as well as differences between different methods are only visible in the tails of the distributions or at hard edges. For $\eta_{\mu2}$ and $\phi_{j2}$, we expect the unfolding to be close to an identity mapping, with hard boundaries. While the DM-networks should be well-suited to those observables, we also find no performance loss for the classic CFM unfolding. Instead, we observe minor deviations of the (C-)SB networks at the hard edges, likely indicating a lack of expressivity in our specific implementation.

Since we turned it into a network input, the di-muon mass $m_{\mu\mu}$ is learned well by all networks. Moving on to complex correlations like the transverse momentum of the first muon, computed from Eq.(53), we still observe excellent agreement. With the exception of (C-)SB networks in the low-$p_T$ region, the precision is at the level of a few per-cent except for fluctuating tails. A known challenge to all generative networks is the $\Delta R_{j1j2}$ distribution. This observable defines a non-trivial derived feature, combining collinear enhancement with a hard

phase cut. All five networks struggle with this feature, and all conditional networks show superior performance.

At the subjet level, the number of jet constituents, $N_{j2}$, tends to be larger at gen-level than at reco-level, since not all particle are eventually detected. This explains the strange peak at zero for $\tau_{3,j_2}$ at reco-level. Here, the jet algorithm clusters less than two particles within one jet, indeed giving $\tau_{3,j_2} = 0$. At gen-level this effect is highly suppressed. The CFM and C-DiDi manage to reliably unfold even this small excess of zeros, while the other networks fit through it. DiDi and the SB deviate above 10% from the truth when getting closer to the tails of the distributions. To compensate the SB is overpopulating the peak region.

## 6  Conclusion

In this paper, we have extended two distribution-mapping, ML-based unfolding methods, SBUnfold and DiDi and benchmarked them with an updated conditional generative unfolding method (CFM). The two distribution-mapping methods have been shown to accurately reproduce the marginal distributions of the target distributions, but were not able to model the correct detector response. By augmenting them with conditioning, C-SB and C-DiDi faithfully reproduce the detector response as well as the marginal target distributions. Like other ML-based unfolding methods, C-SB and C-DiDi are unbinned and readily extend to unfolding in many dimensions simultaneously.

We have started with a detailed discussion of the theoretical foundations of distribution mapping, conditional distribution mapping, and the relation of the SB and DiDi approaches. While this discussion is not needed to understand our results, it allows us to systematically understand the similarities and the benefits of the different generative unfolding architectures. In essence, C-SB and C-DiDi are two realization of the same SDE description, and the main difference between the two implementations is the noise schedule chosen. In addition, within our (C)-SB implementation we sample discrete time steps during training, whereas all other models samples $t$ continuously over a uniform distribution. The impact of conditioning was illustrated simply with a Gaussian example following the mathematical exposition.

Our first benchmark dataset is the well-studied OmniFold/MultiFold dataset composed of six substructure observables. All five networks representing the three methods, classic conditional generation, unconditional DM, and conditional DM, accurately reproduce the distributions in the target unfolding phase space. The two DM-methods are each implemented in Schrödinger Bridge and Direct Diffusion versions. Between our five networks and relative to the truth, the six marginal distributions vary at the per-cent level, with sizable deviations appearing only in the tails of distributions that are consistent with the uncertainty estimate from the Bayesian neural network implementations. The migration analysis for both conditional methods reproduce the physics encoded in the forward simulation. A classifier analysis of the generated unfolded events shows high precision and no clear failure mode for any of the three conditional networks.

Second, we apply generative unfolding to a new, high-dimensional $Z + 2$ jets dataset with 22 phase space dimensions, combining event-level and subjet phase spaces. Again, we find that all networks reproduce the true phase space distribution before the detector with high fidelity. For many of the networks, the one-dimensional marginal distributions agree with the truth at the per-cent level or better, and sizable deviations again appear only in kinematic tails.

With the release of the updated methods, codified in publicly available software, we have added a new ML-based unfolding method to the collider physics toolkit. Just as with classical unfolding, it is critical to have multiple, comparably accurate/precise techniques that have

different methodological assumptions. Distribution mapping is a third type of method that can now be used to compare with standard conditional-generation and likelihood-ratio methods. Further work is required to fully integrate all aspects of a cross section measurement into generative unfolding (e.g. acceptance effects), but the core component (inverting the forward model) is now highly advanced. We look forward to the application of these methods to experimental data in the near future.

## Code and Data availability

The CFM and DiDi codes are available through the Heidelberg hep-ml code and tutorial library, and the Schröedinger bridge code can be found here. The data set used in Section 5 can be found on zenodo.

## Acknowledgements

We thank Lennart Röver for explaining the Fokker-Planck equation to us. AB and TP are supported by the Deutsche Forschungsgemeinschaft (DFG, German Research Foundation) under grant 396021762 – TRR 257 *Particle Physics Phenomenology after the Higgs Discovery*. AB gratefully acknowledges the continuous support from LPNHE, CNRS/IN2P3, Sorbonne Université and Université de Paris Cité. NH and SPS are supported by the BMBF Junior Group *Generative Precision Networks for Particle Physics* (DLR 01IS22079). TP would like to thank the Baden-Württemberg-Stiftung for financing through the program *Internationale Spitzenforschung*, project *Uncertainties – Teaching AI its Limits* (BWST_IF2020-010). VM, SD, and BN are supported by the U.S. Department of Energy (DOE), Office of Science under contract DE-AC02-05CH11231. The authors acknowledge support by the state of Baden-Württemberg through bwHPC and the German Research Foundation (DFG) through grant no INST 39/963-1 FUGG (bwForCluster NEMO). This work was supported by the DFG under Germany's Excellence Strategy EXC 2181/1 - 390900948 *The Heidelberg STRUCTURES Excellence Cluster*. This research used resources of the National Energy Research Scientific Computing Center, a DOE Office of Science User Facility supported by the Office of Science of the U.S. Department of Energy under Contract No. DE-AC02-05CH11231 using NERSC award HEP-ERCAP0021099.

# A Hyperparameters

| Parameter | CFM | DiDi | C-DiDi | SB | C-SB | Classifier |
|---|---|---|---|---|---|---|
| Optimizer | | Adam | | | Adam | Adam |
| Learning rate | | 0.001 | | | 0.001 | 0.001 |
| LR schedule | | Cosine annealing | | | Exponential decay | Cosine annealing |
| Batch size | | 16384 | | | 128 | 128 |
| Epochs | | 300 | | | 20 | 50 |
| Network | | MLP | | | MLP | MLP |
| Number of layers | | 5 | | | 6 | 5 |
| Hidden nodes | | 128 | | | 256 | 256 |
| Dropout | | - | | | - | 0.1 |
| Noise scale | - | 0.1 | 0.1 | 0.1 | 0.1 | - |
| BNN regularization | | 1 | | | - | - |

Table 1: Network and training hyperparameters for all networks trained on the 6d jet substructure dataset. Results are shown in Figs. 4, 5, and 6

| Parameter | CFM | DiDi | C-DiDi | SB | C-SB |
|---|---|---|---|---|---|
| Optimizer | | Adam | | | Adam |
| Learning rate | | 0.001 | | | 0.001 |
| LR schedule | | Cosine annealing | | | Exponential decay |
| Batch size | | 16384 | | | 128 |
| Epochs | 500 | 500 | 2000 | 200 | 200 |
| Network | | Transformer | | | Transformer |
| Embedding dim | | 64 | | | 64 |
| Transformer blocks | | 6 | | | 6 |
| Attention heads | | 4 | | | 4 |
| Feedforward dim | | 256 | | | 256 |
| Noise scale | - | 0.001 | 0.1 | 0.1 | 0.1 |
| BNN regularization | | 1 | | | - |

Table 2: Network and training hyperparameters for all networks trained on the full-dimensional Z+2j dataset. Results shown in Figs. 8

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
