# Peer review of "Generative Unfolding with Distribution Mapping"

_SciPost Physics_

## Round 1 · Referee Report · Anonymous (Referee 1) · 2025-3-28

Strengths

  • It is a very clear presentation of a rather involved subject, with important benefits for the practitioners.
  • Section 2 has a valuable discussion of general aspects of distribution mapping with stochastic differential equations.
  • The proposed methodologies are studied thoroughly, not just by means of one example.
  • Taken together, the toy example and the real-world application make a very compelling case with low-threshold entry points for the reader.

  • The quality of the draft is excellent. Fonts in the body and the figures were made consistent.

  • The code is public.

Weaknesses

Very small editorial remarks:

p2

2nd paragraph: "These results are promising, but due to the ill-posed nature of the problem, it is essential to have alternative methods."
-> It is not clear what is ill-posed here. Please clarify.

last sentence: " ... as well as any unfolding method." -> Please clarify which class this applies to. Clearly, you don't have traditionally binned unfolding in mind here.

p4 There are so few results on unbinned unfolding that it is a stretch to say that Omnifold is a "standard". Please rephrase.

Fig. 8 has too small labels.

Report

This paper draft presents techniques to improve unfolding techniques that traditionally suffer from improperly learning the conditional probabilities in the training data. It fully meets the quality standards of the journal; in fact, I had difficulties in finding any relevant weaknesses.

Requested changes

Please see above.

Recommendation

Publish (surpasses expectations and criteria for this Journal; among top 10%)

---

## Round 1 · Referee Report · Anonymous (Referee 2) · 2025-4-16

Strengths

  1. The migrations maps for 6D OmniFold dataset in Fig. 5 very closely resembles the truth

Weaknesses

  1. No significant weaknesses identified

Report

The authors modified two methods for generative unfolding based on diffusion models, using Schroedinger Bridge and Direct Diffusion, and included additional conditional structure. The results presented in section 4 are quite encouraging, the review of distribution maping, including the presentation of the new formalism the authors developed, is clearly presented in Section 2, including a very illuminating toy example of Gaussian mixture model in Section 3. As such I am very happy to recommend the manuscript for publication in Sci. Post. without any major changes. Below I only include a list to typographical erros.

Requested changes

List of typographical errors:

  • p. 3 : a a new dataset -> a new dataset

  • p. 10: DM-formalism to reproduces -> DM-formalism to reproduce

  • p. 15: unfolded distribution -> unfolded distributions

  • p. 15: the the CFM -> the CFM

  • p. 21: models samples -> models sample

A further note: while conditional distribution mapping (CDM) is defined on p. 11, the abbreviation DM is, I believe, never defined

Recommendation

Publish (surpasses expectations and criteria for this Journal; among top 10%)

---

## Editorial Decision

resubmitted